# Phytochemical Profiling and Biological Activities of *Rhododendron* Subsect. *Ledum*: Discovering the Medicinal Potential of Labrador Tea Species in the Northern Hemisphere

**DOI:** 10.3390/plants13060901

**Published:** 2024-03-21

**Authors:** Martyna Vengrytė, Lina Raudonė

**Affiliations:** 1Laboratory of Biopharmaceutical Research, Institute of Pharmaceutical Technologies, Lithuanian University of Health Sciences, Sukileliu Av. 13, LT-50162 Kaunas, Lithuania; lina.raudone@lsmu.lt; 2Department of Pharmacognosy, Lithuanian University of Health Sciences, Sukileliu Av. 13, LT-50162 Kaunas, Lithuania

**Keywords:** *Rhododendron*, *Ledum*, phytochemical, Ericaceae, pharmacology

## Abstract

*Rhododendron* subsect. *Ledum* is a distinct taxonomic subdivision within the genus *Rhododendron*, comprising a group of evergreen shrubs and small trees. This review will comprehensively analyse the phytochemical profiles and biological properties of the *Rhododendron* subsect. *Ledum* species subsect. *Ledum* consists of eight plant species indigenous to temperate and subarctic regions of the Northern Hemisphere, collectively known as Labrador tea. Recent investigations have concentrated on the phytochemical constituents of these plants due to limited data, emphasizing their evergreen nature and potential industrial significance. This review summarizes their major phytochemical constituents, including flavonoids, phenolic acids, and terpenoids, and discusses their potential biological activities, such as antioxidant, anti-inflammatory, antimicrobial, antitumor, hypoglycemic, hepatoprotective, neuroprotective, and cardioprotective effects. Traditional uses of these plant species align with scientific findings, emphasizing the significance of these plants in traditional medicine. However, despite promising results, gaps exist in our understanding of specific compounds’ therapeutic effects, necessitating further research for comprehensive validation. This review serves as a valuable resource for researchers, identifying current knowledge, uncertainties, and emerging trends in the study of the *Rhododendron* subsect. *Ledum* species.

## 1. Introduction

*Rhododendron* subsect. *Ledum* is a taxonomic subdivision that includes several species of evergreen shrubs and small trees [1]. Belonging to the family Ericaceae, *Ledum* was distinguished as a cluster of eight plant species indigenous to the temperate and subarctic domains of the Northern Hemisphere, which are commonly known as the Labrador tea. Plants of the genus *Ledum* were assigned to the section of the genus *Rhododendron* in 1990 [1,2,3]. Previous *Ledum* species are now accepted as *Rhododendron tomentosum* (Stokes) Harmaja, *Rhododendron groenlandicum* (Oeder) Kron et Judd, and *Rhododendron columbianum* (Piper) Harmaja [4]. These plants are known for their distinct morphological characteristics, such as small leaves and bell-shaped flowers, and they typically thrive in acidic soil habitats [5]. 

In recent years, extensive research has focused on investigating these plants’ phytochemical constituents and biological properties due to a lack of data. *Rhododendron* subsect. *Ledum* species are one of the many evergreen plants, meaning that they retain their leaves throughout the year and into the following growing season [6]. Industrially, the leaves of evergreen plants are an intriguing raw material due to their availability throughout the growing season. However, the phenological cycle of plants is characterized by the chemical heterogeneity of their secondary metabolites [7]. Subsect. *Ledum* species plants are an interesting topic because scientific research is scarce in Europe, and scientific data suggest diverse phytochemical variability throughout their habitats and trends in profile alterations in the frame of the changing climate [8]. Studies in Lithuania show promising results about *R. tomentosum* extract’s phytochemical and biological properties [9]. However, further science-based studies are needed to confirm its application in the medical field. Furthermore, most of the studies done worldwide are mostly about *R. tomentosum*, and there are only a few about *R. groenlandicum* and none about *R. columbianum*. In addition, there is a lack of data on evaluations of the bioactive compounds of these plants, such as proanthocyanidins, flavonoids, phenolic acids, terpenoids, etc. There are no data on the correlation of the phytochemical composition of the subsect. *Ledum* with biological effects. It is known that subsect. *Ledum* species plants accumulate many phytocomponents, showing a broad spectrum of positive effects on health [5]. Likewise, it is also important to know the therapeutic effect appliances of the subsect. *Ledum* plants. The optimization of the separation method and the purification of individual fractions of bioactive compounds are necessary in research on new phytochemical compounds. It is relevant to characterize the chemophenetic profile of this section to substantiate the phytoprofiles of each taxon [10]. All the studies about *Rhododendron* subsect. *Ledum* species are new and relevant since the any obtained functional fractions would open perspectives for developing innovative pharma, food, or divergent products with added value. The aim of this review is to comprehensively examine the current state of phytochemical and biological effect research in the field of subsect. *Ledum* species, analyze key findings, and identify scientific uncertainties, emerging trends, and gaps to provide valuable insights for future investigations.

## 2. Results and Discussion

### 2.1. Nomenclature and Taxonomy

The *Rhododendron* subsect. *Ledum* species plants’ taxonomy is commonly intermingled between old and new names. This problem first occurred when *Ledum* taxa were included in the genus *Rhododendron* in 1990 [1]. As the existing research shows, many authors to this day use old names for *Ledum* species plants: *Rhododendron tomentosum* as *Ledum palustre*, *Rhododendron groenlandicum* as *Ledum latifolium* or *Ledum groenlandicum*, and *Rhododendron columbianum* as *Ledum glandulosum* or *neoglandulosum* [6,11]. There are many different traditional names for *Ledum* species plants, such as Labrador tea, wild rosemary, marsh tea, and marsh rosemary. Somehow, it is hard to tell species apart as, for example, the name Labrador tea is used for all *Ledum* species: *R. tomentosum, R. groenlandicum*, and *R. columbianum*. The only difference is that, in some studies, *R. tomentosum* is called northern Labradors tea, *R. groenlandicum*—bog Labradors tea, and *R. columbianum*—western Labradors tea [6,12,13].

### 2.2. Distribution of the Subsect. Ledum Species

The *Rhododendron* subsect. *Ledum*, a group of plants, can be found in regions in the Northern Hemisphere. These plants have adapted well to challenging conditions. They are commonly found in subarctic and boreal ecosystems. Some key areas rich in species belonging to the *Rhododendron* subsect. *Ledum* include North America and Northern Europe [14]. They typically thrive in wetlands, bogs, and tundra environments. Subsect. *Ledum* species are also distributed across Eurasia, including Russia, Siberia, Scandinavia, and other European countries, except Romania and Great Britain [15]. Furthermore, some types of subsect. *Ledum* can be found in parts of Asia, including the Russian Far East, Japan, and China. The *R. tomentosum* species grows widely in peaty soils in northern and central Europe, the northern part of Asia, and North America [5]. *R. groenlandicum* is specifically located in the northwest and northeast of North America, Greenland, and Canada [16]. *R. columbianum* is widespread in the western United States of America and the western part of Canada [17].

They exhibit their resilience in habitats such as subalpine meadows and acidic peat bogs. These plants are often associated with subalpine environments, demonstrating their ability to adapt to different ecological conditions [18]. Moreover, researchers have been particularly interested in these plants due to their capability to produce a wide range of specialized metabolites, which play a role in their ecological interactions. Their phytochemical profiles can be quite different, elucidating distinct chemotypes and expressing varying biological activities. 

### 2.3. Phytochemical Composition of Rhododendron Subsect. Ledum Species 

*Rhododendron* subsect. *Ledum* species are rich in various secondary metabolites, with flavonoids, phenolic acids, and terpenoids being the major components. Flavonoids, including quercetin, kaempferol, and myricetin derivatives, have been identified as abundant constituents. Phenolic acids, such as gallic acid, ellagic acid, and caffeic acid derivatives, are also prevalent. Terpenoids, including monoterpenes and sesquiterpenes, contribute to the chemical complexity of these plants [8,9,19]. Furthermore, coumarins (fraxetin, fraxin, esculin etc.) and triterpenic compounds (taxerol, uvaol, ursolic acid, sterols) were determined to be present in these plants [5,20,21,22].

#### 2.3.1. Principal Components of Essential Oil in *R. tomentosum*, *R. goenlandicum* and *R. columbianum* Species

Essential oil is located in all plant parts, leaves, shoots, inflorescences, and seeds [5]. Judzentiene et al. conducted a study in Lithuania where it was found that *R. tomentosum* seeds, leaves, and young shoots are rich in essential oil components (47 compounds) such as palustrol, ledol, cyclocolorenone, myrcene, limomene, etc. Conducted research reveals that seeds and shoots of the same plants have different quantities of palustrol and ledol, which are found in the largest amounts in these plants’ compositions. Plant seeds have more palustrol (38.3%) and less ledol (27.0%) in contrast to shoots, where the ledol quantity (36.5%) predominated over palustrol (21.0%). Overall research shows that different parts of the same plant have similar phytochemical compositions, and only quantitative differences are observed [9,23]. Raal et al., in Estonia, analyzed dried *R. tomentosum* shoot samples using gas chromatography–mass spectrometry methods and found 72 compounds. The largest amounts found in the plant’s composition were palustrol (15.9–53.5%), ledol (11.8–18.3%), γ-terpineol (0–31.2%), and p-cymene (0.1–13.9%) (Table 1). In this research, two different chemotypes of *R. tomentosum* were found —for the first, shoots were rich in palustrol (41.0–53.5%) and ledol (14.6–18.3%) and the second one, for the first time, contained more γ-terpineol (24.7–31.2%) and p-cymene (12.5–13.9%) than palustrol (15.9–16.7%) and ledol (11.8–12.8%) [8]. Korpinen et al., in 2021, determined palustrol, ledol, and β-myrcene to be principal compounds in the stems and leaves of *R. tomentosum* Table 1 [24]. Indeed, the plants growing in central and northern Europe belong to the palustrol-ledol chemotype. The trend of the higher percentage of palustrol can also be observed towards the northern latitudes of European growing areas. However, extensive phytogeographical research is needed to elucidate the essential oil geochemotypes. Ascaridole and p-cymene (64.7% and 21.1%) were the predominant compounds in the essential oil of leaves of *R. subarcticum* grown in Canada [25]. This species’ name is now regarded as a synonym for *R. tomentosum* [26]. Indeed, up to date data in the literature suggest about 10 chemotypes of *R. tomentosum*, determined all over the Northern Hemisphere [27]. The *R. tomentosum* materials of precisely defined chemotype and phytochemical markers could be applied for specific biological properties and consistent therapeutic effects. Essential oil of *Rhododendron tomentosum* ssp. *subarcticum* growing in controlled conditions in Alaska demonstrated cymene and α-pinene as its principal compounds, with amounts varying from 35 to 56% and 2 to 7%, respectively [18]. *R. tomentosum* (originated form Miszewko, Poland) cultures propagated in cultures showed an essential oil profile with predominant components of alloaromadendrene and p-cymene, while maternal plants from natural bogs showed a prevalence of palustrol and ledol [28,29]. 

Research from Canada shows the composition of various essential oil compounds in native *R. groenlandicum* [30,31]. Over one hundred and sixty different compounds were determined. The basis of its phytochemical composition, compared to *R. tomentosum,* is quite similar. Great variability is observed in the percentages of compounds such as limonene, p-cymene, sabinene, and cis- and trans-p-mentha-1,7(8)-diene. Major differences occur in palustrol and ledol in *R. groenlandicum*, although these are very minor compounds (<1%) in opposition to what is observed in *R. tomentosum* [30]. Furthermore, Lagha et al. determined α- and β-selinene (19.8%), sabinene (11.9%), germacrene (11.6%), and germacrone (8.5%) as principal essential oil components [31]. Etienne determined α-selinene, germacrene B, and α-pinene as prevailing components, accounting for 13.99%, 13.20%, and 8.59%, respectively [32]. Indeed, these variations in essential oil compositions can be driven by environmental factors, genotypes, and adaptation to specific climatic, edaphic, and ecological conditions [18,33]. 

*R. tomentosum* and *R. groenlandicum* are the main species with ethnopharmacological data and several scientific papers elucidating their essential oil chemotype variabilities [6]. *R. groenlandicum* has a great ethnopharmacological history in Canadian Cree nations, with records of anti-inflammatory activities towards the liver, kidney, lung, and digestive system diseases [34,35]. The main determined chemotypes of Canadian-origin *R. groenlandicum* are the sabinene—limonene and sabinene—and β-selinene chemotypes (Figure 1) [30,31]. On the other hand, *R. tomentosum*, in traditional medicinal systems, was used for skin, rheumatic, and arthritic diseases, expressing its diuretic and analgetic activities [34,35]. For the Canadian chemotype of *R. tomentosum (R. tomentosum* ssp. *Subarcticum)*, the main essential oil compounds are ascaridole and p-cymene, while the European species mainly represents ledol and palustrol chemotypes in different geography-dependent ratios (Figure 1) [9,23,24,25,36]. Nevertheless, there are data of European *R. tomentosum* for which the essential oil composition mainly contains p-cymene and isoascaridole (Figure 1) [37].

Uncertainties occur with the accepted species *R. columbianum*, which has many synonyms such as *R. glandulosum*, *R. neoglandulosum*, and others [38]. Still, to the best of our knowledge, no literature data are found regarding the chemical composition of this species. Moreover, the ethnopharmacological data are also lacking. 

**Table 1 plants-13-00901-t001:** Essential oil composition in *Rhododendron* subsect. *Ledum* species.

Components	Plant Part	Extraction Method	Species	Detection Method	Country	Reference	Bio-Activity
Palustrol (15.9–53.5%),	Plant shoots	hydrodistillation	*R. tomentosum*	Gas chromatography—mass spectrometry methods	Estonia	[8]	_
Ledol (11.8–18.3%),							
γ-terpineol (0–31.2%),							
p-cymene (0.1–13.9%)							
Ledol (36.5%),	Plant shoots	hydrodistillation	*R. tomentosum*	Gas chromatography—mass spectrometry methods	Lithuania	[9]	_
Palustrol (21.0%),							
Ascadirole (4.0%),							
Lepanone (3.0%),							
Lepanol (2.8%),							
P-cymene (2.2%),							
Myrcene (1.9%)							
Palustrol (38.3%),	Plant seeds	hydrodistillation	*R. tomentosum*	Gas chromatography—mass spectrometry methods	Lithuania	[9]	_
Ledol (27.0%),							
P-Cymene (1.7%), Lepalol (1.6%),							
Geraniol (1.2%)							
Palustrol (24.6–33.5%)	Plant shoots and inflorescences	hydrodistillation	*R. tomentosum*	Gas chromatography—mass spectrometry methods	Lithuania	[9]	Antioxidant activity, Antifungal activity against Candida Parapsilosis
Ledol (18.0–29.0%)							
Ascadirole (7.0–14.0%),							
Myrcene (7.2–10.1%)							
Lepanol (3.3–7.9%)							
cyclocolorenone isomers (4.1%)							
β-myrcene (31%)	Plant stems and leaves	Hydrodistillation	*R. tomentosum*	Gas chromatography—mass spectrometry methods	Finland	[24]	_
Palustrol (38.8%)							
Ledol (15.9 %)							
Sabinene (0.05–35.0%)	Plant stems and leaves	hydrodiffusion	*R. groenlandicum*	Gas chromatography—flame ionisation detector (GC-FID)	Canada	[30]	_
β-pinene (0.05–8,4%)							
*p*-cymene *(0.2–3.4%)*							
Limonene (0.3–67.0%)							
Camphene (1.3%)							
a-terpinene (2.3%)							
Terpinolene (1.5%)							
Terpinen-4-Ol (0.5–5.1%)							
Myrtenal (0.3–3.8%)							
Bornyl acetate (0.3–8.4%)							
Sabinene (11.93%)	Plant leaves	-	*R. groenlandicum*	Gas chromatography–mass spectrometry (GC–MS) and gas chromatography/flame-ionization detection (GC/FID)	Canada	[31]	Antibacterial activity
β-Selinene (10.95%)							
Germacrene B (9.75%)							
α-Selinene (8.89%)							
Germacrone (8.51%)							
Ascaridole (67.7%)	Plant leaves	hydrodistillation	*Rhododendron tomentosum* ssp. *subarcticum*	Gas chromatography–mass spectrometry (GC–MS) and gas chromatography/flame-ionization detection (GC/FID)	Canada	[25]	Antiparasitic activity
p-cymene (21.1%)							
Terpinen-4-ol (2.5%)							
β-pinene (1.2%)							
Sabinene (17.9%)	Plant stems and leaves	hydrodistillation	*R. tomentosum*	Gas chromatography–mass spectrometry (GC–MS)	Korea	[39]	Antioxidant activity, antimicrobial activity,
Terpinen-ol (7.61%)							
Myrtenal (7.44%)							
β-selinene (6.5%)							
Myrtenol (3.53%)							
p-cymene (25.5%)	Plant stems and leaves	hydrodistillation	*R. tomentosum*	Gas chromatography–mass spectrometry (GC–MS)	Poland	[37]	Insecticidal activity
Isoascaridole (20.5%)							
is-ascaridole (14.8%)							
Geranyl acetate (4.2%)							

#### 2.3.2. Phenolic Compounds in *R. tomentosum*, *R. goenlandicum* and *R. columbianum* Species

*R. tomentosum*’s flavonoids are mainly composed of flavonol glycosides, namely quercitrin, isoquercitrin, hyperoside, rutin, methylated, and phenolic acid substituted quercetin and myricetin compounds [Figure 2]. The second rich fraction is flavan-3-ols with the prevailing compounds catechin, epicatechin, and B-type proanthocyanidins [5,40,41,42]. Dufour et al., in 2007, determined that leaves and twigs of *R. groenlandicum* contained 20 and 39 g/100 g of total phenolic compounds, respectively [35]. The fraction of *R. groenlandicum* with adipogenic activity was characterized by catechin, chlorogenic acid, epicatechin, quercetin-3-*O*-galactoside, quercetin-3-*O*-glucoside, quercetin-glycoside, quercetin-3-*O*-arabinoside, and quercetin as the key components of its phenolic profile [Figure 2] [36]. Black et al., in 2011, determined a significant seasonal variation in individual phenolic compounds, with the greatest amounts being observed at the end of active vegetation period [41]. Rapinski et al. determined 14 phenolic-origin compounds, with quercetin derivatives and flavan-3-ol being the dominant ones. Their study supports the interplay between distinct chemotypes in the boreal forest, taiga, and tundra ecosystems in the Quebec region [43]. The phenolic acids profile contains chlorogenic acid and a body of caffeic acid substituted derivatives [9,41,43,44] Table 2. 

The main research on phenolic profiling was performed on species (*R. tomentosum* ssp. *subarticum* and *R. glandulosum*) grown in Canadian regions in regards to ethopharmacological significance [30,31,34,35,36,43]. Key differences between the phenolic profiles in the latter two species were within the profile of proanthocyanidins and the presence of taxifolin derivatives [44]. *Rhododendron* species contain notable amounts of proanthocyanidins [36,41,42,43,44]; nevertheless, information regarding subsect. *Ledum* species remains scarce. Research on proanthocyanidins and their functionalization spans the fields of ecology, medicine, materials science, and agriculture, making it a multidisciplinary field with significant implications for solving scientific uncertainties and practical applications.

In light of the current state of research, it is evident that data on phenolic compounds in *Rhododendron* subsect. *Ledum* species are limited, and there remains a notable scarcity of data addressing the quantitative and qualitative variations in these compounds throughout the vegetation cycle and across diverse geographical locations. Further research is needed to elucidate the intricate interplay of environmental factors and biological processes shaping the phenolic composition of these plants, offering valuable insights for both scientific understanding and potential applications in various fields.

#### 2.3.3. Triterpenic Compounds in *R. tomentosum*, *R. goenlandicum* and *R. columbianum* Species

The *Ledum* species *L. palustre* and *L. grenlandicum* contain triterpenic compounds, namely ursolic acid, uvaol, and uvaol acetate (Figure 3) [20]. Dufour et al., in 2007, were the first to identify ursolic acid in the twigs of *Ledum goenlandicum* (*R. groenlandicum*) and demonstrated its rich fraction of anticancer activity against DLD-1 and A-549 cell lines [40]. The content of ursolic acid in the ethanolic extract of aerial parts of *R. tomentosum* was 15.2 mg/g [33]. *L. palustre* (*R. tomentosum*) leaves from China contained ledumone, uvaol, lepenone, α-amyrenone, ursolic acid, lupeol, α-amyrin, fern-9(11)-ene-2α,3β-diol, and fernenol (Figure 3) [42].

#### 2.3.4. Other Compounds in *R. tomentosum*, *R. goenlandicum* and *R. columbianum* Species

Shotyk et al. determined trace minerals in *R. groenlandicum* leaves. Study shows that the most abundant trace mineral in leaves is manganese (±706 mg/kg), with a little bit smaller amounts of aluminum, iron, zinc, copper, and nickel. The least prominent one is silver (± 0.57 g/kg) and other heavy metals. As shown in existing research, *R. groenlandicum* tends to accumulate essential trace minerals rather than potentially toxic heavy metals [22]. Wang et al. determined the stilbene compound polydatin and coumarin derivatives in the leaf samples of *R. tomentosum* [47].

### 2.4. Biological Activities of Rhododendron Subsect. Ledum Species

Subsect. *Ledum* plants share common names, which can lead to misidentification and potentially harmful consequences. Labrador tea is the common name under which, in various sources in the literature, *R. tomentosum*, *R. goeanlandicum*, and *R. columbianum* are indicated [4,6]. Each species can also have a specific common name, such as northern Labrador tea, bog Labrador tea, and western Labrador tea or trapper’s tea for *R. tomentosum*, *R. goeanlandicum*, and *R. columbianum*, respectively. Labrador tea has a history of traditional use by Indigenous peoples in North America. It has been used for various medicinal purposes, such as treating colds, respiratory issues, digestive problems, and more (Table 3). It is also used as a beverage tea. Bog Labrador tea has similar uses, such as treating colds and digestive issues. It is also known for its pleasant aroma and is sometimes used as a tea beverage. Common names for plants can vary greatly from one region or culture to another, leading to confusion. Scientific names, on the other hand, provide a universal and standardized way to identify a particular plant species. This precision is crucial in validating biological activities [48].

The subsect. *Ledum* plants show a large amount of medical and therapeutic potential that could be used in future science-based evidence research. The phytochemical diversity of *Rhododendron* subsect. *Ledum* species is associated with a broad spectrum of biological activities. Traditionally, these plants were used against inflammation, pain, skin ailments, the common cold, and gastrointestinal disorders, as well as being used as repellents [49]. Studies have shown that these plants possess antioxidant properties, allowing them to scavenge free radicals and mitigate oxidative stress [5]. They also exhibit anti-inflammatory effects by modulating pro-inflammatory mediators and signaling pathways. Additionally, *Rhododendron* subsect. *Ledum* species demonstrate antimicrobial activity against various pathogens and potential antitumor, hepatoprotective, neuroprotective, and cardioprotective effects [5]. The significant biological activities observed in the *Rhododendron* subsect. *Ledum* plants suggest their potential medicinal and therapeutic applications. Their antioxidant and anti-inflammatory properties make them attractive candidates for the prevention and treatment of oxidative stress-related disorders, including cardiovascular diseases, neurodegenerative conditions, and cancer [5,41]. Moreover, their antimicrobial activity opens possibilities for developing novel antimicrobial agents [39,50]. Furthermore, *Rhododendron* subsect. *Ledum* species exhibit hepatoprotective effects, which may have activity for liver health [5]. Numerous epidemiological and clinical studies have reported health improvements associated with antioxidant intake: decreased prevalence of cancer, increased memory function, increased physical endurance capacity, and cardioprotection [41]. *R. tomentosum* is considered to be an effective antioxidant for preventing and treating free-radical pathologic conditions, such as chronic bronchitis and asthma [51]. *R. tomentosum* extracts have been proven to protect animals from the injury to both the gastrointestinal tract and the hemopoietic system [52]. *R. tomentosum* was evaluated as a source of potential antiarthritic drugs. It has anti-proliferative and pro-apoptotic activities that are beneficial for rheumatoid arthritis treatment [26]. Likewise, some populations have a strong consensus for selected usage of *R. tomentosum* for stomachache, cold symptoms, and toothache [53]. Another study in vitro verified the antidiabetic activity of some herbs used in the traditional medicine of the Cree community from northern Quebec in Canada, revealing that *R. tomentosum* has an interesting antidiabetic potential [54].

**Table 3 plants-13-00901-t003:** Ethnopharmacological uses of *R. tomentosum*.

Region	Part Used	Ethnopharmacological Uses	Preparation	References
Asia and northern europe	herb (leaves)	Arthrosis, Rheumatism, bronchitis, lung diseases, bug bites, pain relief, wounds, itch, eruptions, cold and fever, cough, sore throat, dyspepsia, dysentery, gout, leprosy and whitlow	–	[50,55]

Estonia	herb (leaves)	coughs, tuberculosis, cold, rheumatic diseases	–	[56,57]
Estonia	dried branches	repellent against bedbugs, clothing moths, fleas	–	[57]
Sweden, norway,	herb (leaves)	against lice	–	[57]
Finland				
Norway, denmark	herb (leaves)	cold, whooping cough, for rheymatism as pain reliever, for high blood pressure, bladder catarrh and diphtheria	–	[58]

Russia	herb (leaves)	bronchitis, tuberculosis, cough, asthma, spastic enterocolitis	the infusion	[5]

	herb (leaves)	as anthelmintic, fever, urethritis, metrorrhagia, women’s diseases and gastritis.	the decoction	[5]

	herb (leaves)	eczema, scabies, insect stings, bruises, wounds, boils, hematomas, ringworm, chicken pox, blepharitis and conjuctivitis.	the oinment on the base of linseed oil or animal fats	[5]


	herb (leaves)	Rhinitis	drops	[5]
	herb (leaves)	Hypnotic and sedative effect	by smoking	[5]
Yakutia (russia) and bulgaria	herb (leaves)	as abortifacient	–	[5]
Poland	herb (leaves)	Toothache and painful gums	mouth rinsing solution	[59]
China	Leaves	Infection and inflammation	–	[50]
Korea	Leaves	Female disorders	–	[50]
Tibet (china)	herb (leaves)	tuberculosis, bronchitis, endometris, jaundice and liver disease	the infusion and decoction	[5]

	herb (leaves)	gynecological diseases	bath form	[5]
	herb (leaves)	diarrhea	in the form as ash	[5]

#### 2.4.1. Antioxidant Properties

The antioxidant properties of *Rhododendron* subsect. *Ledum* are primarily attributed to its flavonoids and phenolic acids [5]. These compounds neutralize reactive oxygen species (ROS) and inhibit oxidative damage to cellular components. They do so by scavenging free radicals, chelating metal ions, and modulating antioxidant enzyme activities, including superoxide dismutase (SOD) and catalase [39,60].

#### 2.4.2. Anti-Inflammatory Properties

*Ledum’s* anti-inflammatory effects are mediated through the regulation of key molecular pathways. Flavonoids and terpenoids found in *Ledum* have been shown to inhibit pro-inflammatory cytokines such as interleukin-6 (IL-6) and tumor necrosis factor-alpha (TNF-α) [27]. Additionally, they can modulate the activity of nuclear factor-kappa B (NF-κB), a central regulator of inflammation. There was an in vitro study conducted in which the anti-inflammatory activity of *R. tomentosum* extract verifyied the traditional use of this herb as a painkiller. The moderate inhibition of prostaglandin biosynthesis and PAF-induced exocytosis was obtained [54,61]. The anticancer activity of *R. tomentosum* was examined with in vitro and in vivo studies. Two quercetin glycoside derivatives isolated from leaves—quercetin 3-β- D-(6-p-coumaroyl) galactoside and quercetin 3-β-D-(6-p- hydroxy-benzoyl) galactosid—were found to be cytotoxic against human mouth epidermal carcinoma [50].

#### 2.4.3. Antimicrobial and Antiviral Properties

The antimicrobial activity of *Ledum* compounds involves the disruption of microbial cell membranes, inhibition of essential enzymes, and interference with bacterial and fungal cell wall synthesis. Collectively, these mechanisms lead to microbial growth inhibition and cell death. The antifungal activity of quercetin 3-β-D-(6-p-coumaroyl) galactoside and quercetin 3-β-D-(6-p-hydroxy-benzoyl), isolated from leaves of *R. tomentosum*, was established in vitro by the micro-broth dilution method [5]. In addition, *R. tomentosum* (50 μg/mL) exhibited an effect on the basal and insulin-stimulated 3H-deoxy-glucose uptake in differentiated 3T3-L1 adipocytes [54]. Another study shows that hydro-distillated stems and leaves essential oil from *R. tomentosum* exhibits insecticidal activity on mosquitoes, moths, and flies in vivo. The essential oil was effective against *Culex quinque—fasciatus, Spodoptera littoralis*, and *Musca domestica*. It was not toxic to non-target *Eisenia fetida* earthworms and moderately toxic to *Daphnia magna* microcrustaceans, over the positive control α-cypermethrin [37]. *R. tomentosum* is known for its antifungal properties against *Cryptococcus neoformans, Saccharomyces cerevisiae*, and *Aspergillus niger* [50]. Antimicrobial action was also observed against *Streptococcus pneumoniae, Clostridium perfringens, Mycobacterium smegmatis*, and *Acinetobacter lwoffii* [39]. Data in the literature also mention the bacteriostatic activity of *R. tomentosum* sprouts against *Mycobacterium tuberculosis* isolated from drug-resistant and drug-sensitive patients with active pulmonary tuberculosis forms. In vivo studies with rodents showed that the infusion of this plant can enhance the activity of antituberculous drugs [50]. An in vivo study was performed on the pharmacokinetics of aesculin, aesculetin, fraxetin, fraxin, and polydatin in canine plasma. The pharmacokinetic analysis revealed varying plasma concentrations, absorption rates, and metabolic fates of the identified coumarine derivatives, following the oral administration of *R. tomentosum* extract [47]. Plant extracts contain bioactive compounds that interact with the composition and activity of the human microbiome. These modifications in the microbiome could contribute to improved digestion, enhanced nutrient absorption, and strengthened immune function [62]. Wang et al. emphasized the application of combined analytical systems enabling future investigations into the effects of *R. tomentosum* (*L. palustre*) extract on the microbiome and metabolism [47].

#### 2.4.4. Antidiabetic Properties

There were interesting results shown in an adipogenesis assay in vitro using 80% ethanol extract of *R. groenlandicum*, which was able to stimulate adipogenesis to a similar extent as rosiglitazone, a representative TZD oral hypoglycemic [63]. *R. groenlandicum* is also one of the subsect. *Ledum* species plants which show potential for use in future therapeutic profiles. There are studies showing that *R. groenlandicum* extract exerts benefits in restoring glucose homeostatic mechanisms in mice fed with a high-fat diet [64]. Furthermore, *R. groenlandicum* reduced blood glucose and insulin while improving the response to an oral glucose tolerance test (OGTT) in in vivo studies with mice [64]. Eid et al., in 2016, determined that catechin and epicatechin, in combination, could be the key compounds responsible for the adipogenic activity of Labrador tea crude ethanolic leaf extract [36]. *R. groenlandicum* treatment improves microalbuminuria and significantly reduces renal fibrosis and steatosis [64]. Indeed, these studies show an evident potential of *R. groenlandicum* for use in the future treatment of diabetes. 

#### 2.4.5. Anti-Cancer Properties

The anti-cancer potential of *Rhododendron* subsect. *Ledum* is still under investigation, but it is believed to involve various mechanisms. Flavonoids and phenolic acids may induce apoptosis (programmed cell death) in cancer cells, inhibit cell proliferation, and disrupt angiogenesis, the process by which tumors develop their own blood supply, as results have shown in in vitro and in vivo studies with mice [46]. 

#### 2.4.6. Other

Due to the lack of data and research on *R. columbianum*’s phytochemical and bioactive properties, there is no information about this plant’s therapeutic effects.

### 2.5. Rhododendron Subsect. Ledum Species Toxicity

*R. tomentosum* is regarded as a poisonous plant due to the content of toxic volatile compounds, especially sesquiterpenoid ledol, in its essential oil [5,6,23]. Although a low concentration of ledol in the beverage may have a restorative effect similar to caffeine, large doses can affect the central nervous system. Initially, psychomotor stimulation occurs, afterwards seizures and cramps, and eventually paralysis, breathing problems, and even death [6]. However, the absence of clinical evidence means that there are no data available regarding the safe dosage of *R. tomentosum*. Judzentiene et al. conducted an in vivo study of *R. tomentosum* inflorescence and shoot essential oils’ toxicity, using brine shrimp *Artemia* sp. (larvae). The study revealed that all samples were notably toxic. Essential oils obtained from shoots gathered in September (seed-ripening stage) that contained appreciable amounts of palustrol (26.0 ± 2.5%), ledol (21.5 ± 4.0%), and ascaridol (7.0 ± 2.4%) appeared to be the most toxic (Figure 1) [23].

According to research and ethnopharmacological data, *R. groenlandicum* is less toxic than *R. tomentosum* because of the minimal ledol quantity in its essential oil composition [5,6,58]. 

The literature shows another major toxic compound which appears to be found in *Rhododendron* species leaves—grayanotoxin I (Figure 4) [65]. It is a cyclic diterpene with biological activity similar to the *Veratrum* alkaloids. The symptoms of poisoning include dizziness, hypotension, vomiting, lack of coordination, and, finally, progressive paralysis. Grayanotoxin intoxication is usually related to the consumption of contaminated honey, called “mad honey disease”, which is produced by bees from *Rhododendron* species nectar [6]. However, there is only one article from the 19th century which declares that *R. tomentosum* is free of grayanotoxins [66]. There is no information about grayanotoxins in other *Ledum* plants—such as *R. groenlandicum* and *R. columbianum*. Therefore, as detailed in this study, there are a lack of new extensive studies of *Rhododendron* subsect. *Ledum* plants’ toxicity, especially grayanotoxins’ appearance in compositions. 

## 3. Materials and Methods

The literature on nomenclature, botany, phytochemistry and biological activities of *Rhododendron* Subsect. *Ledum* was collected using keywords such as accepted plants names: “*Rhododendron tomentosum*”, “*Rhododendron groenlandicum*”, and “*Rhododendron columbianum*”; synonyms: “*Ledum palustre* ”, “*Ledum latifolium*”, “*Ledum groenlandicum*”, “*Ledum glandulosum*”, and “*Ledum neoglandulosum*”; common names: “Labradors tea”, “wild rosemary”, “marsh tea”, and “marsh rosemary”; “Phytochemistry”; etc. Data were collected from scientific databases including Google Scholar, Pubmed, Springer link, and ScienceDirect. A total of 67 articles were identified, most of which were related to phytochemistry, ethnopharmacology, and biological activities. 

## 4. Conclusions

Although the findings regarding the phytochemical profiles and biological activities of *Rhododendron* subsect. *Ledum* species are promising, further research is needed to explore their potential fully. Current data suggest that the phytogeographical and ecosystematic chemotypes of *R. tomentosum* and *R. groenlandicum* are especially promising. However, the body of common names suggests the intermingled usage of various subsect. *Ledum* species. Therefore, research on phenolic, triterpenic, and coumarin compounds could be subjected towards interspecific fingerprinting and elucidating phytochemical markers for each species. Proper plant identification is a prerequisite for consistent studies, ensuring the further functionalisation of subsect. *Ledum* species. Subsect. *Ledum* plants contain a wide range of specialized metabolites that are still underutilized. Comprehensive studies on these compounds’ bioavailability and toxicity profiles are essential for human consumption or medicinal use to ensure the safety and efficacy of potential pharmaceutical applications. Sustainable harvesting and cultivation practices should also be emphasized to ensure conservation in natural distribution areas. Furthermore, there is a significant gap in our knowledge regarding *R. columbianum*. Future research should prioritize investigations into this lesser-researched species’ phytochemical composition, biological activities, and ecological significance. This will contribute to a more comprehensive understanding of *Rhododendron* subsect. *Ledum*.

## Figures and Tables

**Figure 1 plants-13-00901-f001:**
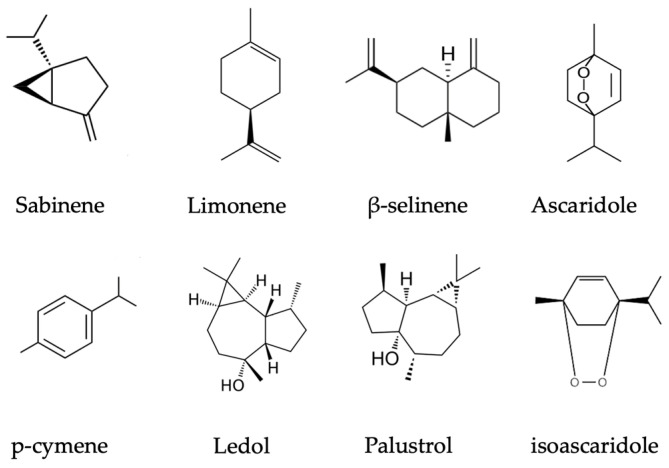
*Rhododendron* subsect. *Ledum* species chemotypes’ volatile chemical contituents.

**Figure 2 plants-13-00901-f002:**
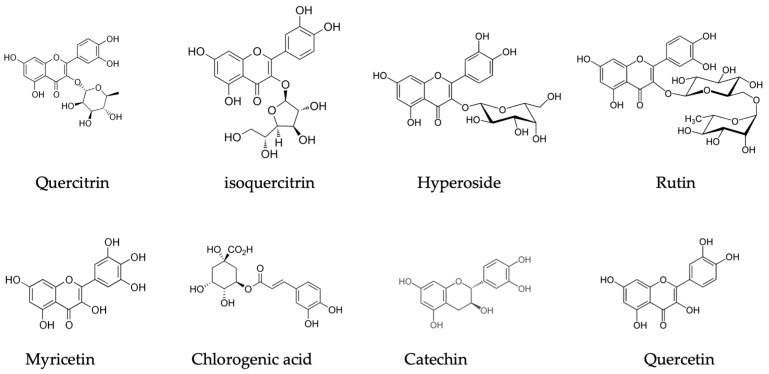
*Rhododendron* subsect. *Ledum* species major phenolic compounds.

**Figure 3 plants-13-00901-f003:**
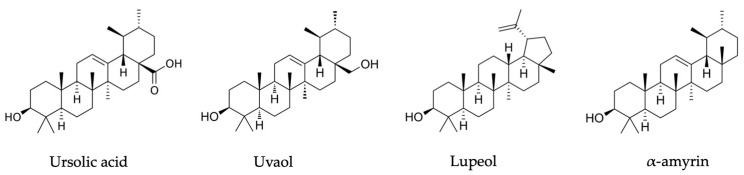
*Rhododendron* subsect. *Ledum* species major triterpenic compounds.

**Figure 4 plants-13-00901-f004:**
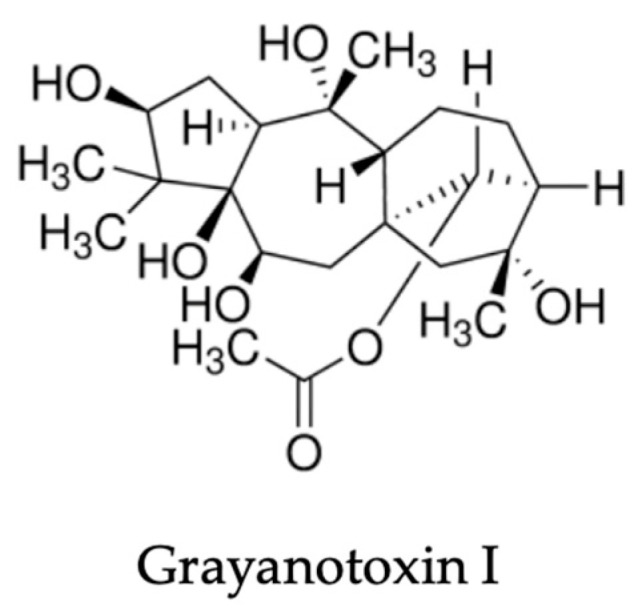
Grayanotoxin I structure.

**Table 2 plants-13-00901-t002:** Phenolic and triterpenic composition of *Rhododendron* subsect. *Ledum* species.

Components	Plant Part	Extraction Method	Species	Detection Method	Country	Reference	Bio-Activity
(+)-catechin	twigs	80% ethanol	*Rhododendron tomentosum* ssp. *subarcticum*	HPLC-DAD	Canada	[41]	Antioxidant activity; TNF-α anti-inflammatory
Quercetin pentoside							
Quercetin 3-*O*-galactoside (4.58 mg/gDW)							
procyanidin B2							
procyanidin B3							
Procyanidin B1							
Caffeic acid derivatives							
Myricetin, quercetin, quercetin 3-*O*-glucoside, quercetin 3-*O*-rhamnoside—minor compounds							
quercetin-3-galactoside	leaves	80% ethanol	*Rhododendron groenlandicum*	HPLC-DAD	Canada	[43]	—
quercetin-glycoside							
(+)-catechin							
chlorogenic acid							
(–)-epicatechin							
Taxifolin glycoside	leaves	80% ethanol	*Rhododendron tomentosum* ssp. *subarcticum*	HPLC-DAD	Canada	[44]	—
taxifolin							
Catechin							
Chlorogenic acid							
(−)-Epicatechin							
(+)-Catechin							
Caffeoylquinic acid							
Proanthocyanidin A1							
Quercetin-3-*O*-galactoside							
Quercetin-3-*O*-glucoside							
Proanthocyanidin A2							
Quercetin glycoside							
Myricetin							
(+)-Catechin	leaves	80% ethanol	*Rhododendron groenlandicum*	HPLC-DAD	Canada	[44]	-
Chlorogenic acid							
(−)-Epicatechin							
Caffeoylquinic acid							
Procyanidin B2							
Procyanidin A1							
Quercetin-3-*O*-galactoside							
Quercetin-3-*O*-glucoside							
Proanthocyanidin A2							
Quercetin glycoside							
Myricetin							
(+)-Catechin	leaves	80% ethanol	*Rhododendron groenlandicum*	HPLC-DAD-MS	Canada	[36]	-
(−)-Epicatechin							
Quercetin							
Chlorogenic acid							
Quercetin-3-*O*-galactoside							
Quercetin-3-*O*-glucoside							
Quercetin-3-*O*-arabinoside							
quercetin-glycoside							
(+)-Catechin	leaves	80% methanol	*Rhododendron groenlandicum*	UHPLC-PDA	Canada	[42]	-
(−)-Epicatechin							
Procyanidin A1							
Proanthocyanidin A2							
Quercetin-3-*O*-galactoside							
Quercetin-3-*O*-glucoside							
Quercetin glycosides							
Quercetin							
Myricetin derivatives							
Myricetin							
uvaol	leaves	chromatographic purification	*Rhododendron tomentosum (Ledum palustre)*	HPLC	China	[45]	-
lepenone							
α-amyrenone							
ursolic acid							
lupeol							
amyrin							
α-fern-9(11)-ene-2α,3β-diol							
fernenol							
6a-hydroxy-14-taraxerene-3,16,21- trione	leaves	absolute ethanol	*Rhododendron tomentosum (Ledum palustre)*	HR-ESI-TOFMS	China	[45]	-
6a,26-dihydroxy-14-taraxerene-3,16,21-trione							
ursolic acid	leaves	acetone	*Rhododendron tomentosum*	GC-MS	Alaska	[46]	anti -acute myeloid leukemia activity.
aesculin	leaves	ethanol	*Rhododendron tomentosum*	UPLC-MS; HPLC	China	[47]	anti-inflammatory, anti-oxidant, anti-tumor, anti-viral
aesculetin							
fraxetin							
fraxin							
polydatin							
Chlorogenic acid	leaves	decoctions with boiling water; ethanol	*Rhododendron groenlandicum*	HPLC-DAD	Canada	[34]	inhibition towards CYP3A4
(+)-Catechin							
(−)-Epicatechin							
Quercetin-3-*O*-galactoside							
Quercetin-3-*O*-rutinoside							

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
