# Peer review of "Phytochemical Profiling and Biological Activities of *Rhododendron* Subsect. *Ledum*: Discovering the Medicinal Potential of Labrador Tea Species in the Northern Hemisphere"

_plants, 2024, doi:10.3390/plants13060901_

Round 1

Reviewer 1 Report

Comments and Suggestions for Authors

The review on Phytochemical Profiling and Biological Activities of Rhododendron Subsect. Ledum summarizes literature review of the medicinal potential of three Labrador Tea Species.

 According to this review, the major phytochemical constituents, includes flavonoids, phenolic acids, terpenoids, and it discusses their potential biological activities, such as antioxidant, anti-inflammatory, antimicrobial, antitumor, hypoglycemic, hepatoprotective, neuroprotective, and cardioprotective effects. Traditional uses align with scientific findings, emphasizing the significance of these plants in traditional medicine.

 This review serves as a valuable resource for scientists regarding current knowledge in the study of the Rhododendron subsect. Ledum species.

Here are some comments:

Table 1 is very helpful. I am suggesting to add 2 more Tables as described below:

1.     Phytochemical composition: Page 3, Section 2.3.1. 

Add a Table with compounds name and % of each chemical obtained from the literature. Include plane parts, extraction solvents, methods of detection if available in the literature, and references. This will be very helpful.

 2.     If possible, Authors can make another Table like Table 1, with the biological properties and extract solvents from various plant parts (leaves, seeds etc.) with reaction mechanism, if they are published. Example: Antimicrobial activity: In vitro or in vivo

                              Extraction solvent

                              Part of the plant (leaves, etc.)

                              Mechanism of action

                              Studied on which species (bacteria, fungus, insects, etc.)

                              Corresponding References

Like Table 1, this Table will be helpful for the researchers when they will plan the study.

 3.     Which studies were done in animals and which studies were from human should be clarified.

4.  In vitro studies : add more detail if available in the literature

Author Response

We are pleased to resubmit for publication the revised version of the manuscript “Phytochemical Profiling and Biological Activities of Rhododendron Subsect. Ledum: Discovering the Medicinal Potential of Labrador Tea Species in the Northern Hemisphere” (Manuscript ID:  plants-2858667). We have appreciated the reviewer 1 comments and have addressed their concerns as outlined below.

Reviewer 1: The review on Phytochemical Profiling and Biological Activities of Rhododendron Subsect. Ledum summarizes literature review of the medicinal potential of three Labrador Tea Species. According to this review, the major phytochemical constituents, includes flavonoids, phenolic acids, terpenoids, and it discusses their potential biological activities, such as antioxidant, anti-inflammatory, antimicrobial, antitumor, hypoglycemic, hepatoprotective, neuroprotective, and cardioprotective effects. Traditional uses align with scientific findings, emphasizing the significance of these plants in traditional medicine. This review serves as a valuable resource for scientists regarding current knowledge in the study of the Rhododendron subsect. Ledum species.

Here are some comments:

Table 1 is very helpful. I am suggesting to add 2 more Tables as described below:

1.     Phytochemical composition: Page 3, Section 2.3.1. 

Add a Table with compounds name and % of each chemical obtained from the literature. Include plane parts, extraction solvents, methods of detection if available in the literature, and references. This will be very helpful.

Answer: Thank you for your constructive feedback. We appreciate your suggestion and included tables on the essential oil composition, phenolic and triterpenic composition. We have included the information regarding the solvent, plant part, geographical origin and bioactivity where available.

 2.     If possible, Authors can make another Table like Table 1, with the biological properties and extract solvents from various plant parts (leaves, seeds etc.) with reaction mechanism, if they are published. Example: Antimicrobial activity: In vitro or in vivo

                              Extraction solvent

                              Part of the plant (leaves, etc.)

                              Mechanism of action

                              Studied on which species (bacteria, fungus, insects, etc.) 

                              Corresponding References

Like Table 1, this Table will be helpful for the researchers when they will plan the study.

Answer: Thank you very much. We constructed tables 1 and 2 with the volatile and nonvolatile compounds determined and indicated the available information regarding species, parts, extraction and detection methods and activities where available. The tables present the current data available in regard to phytogeography and elucidate gaps for further research.

 3.     Which studies were done in animals and which studies were from human should be clarified. 

Answer: The information regarding the experimental mode was added in the manuscript indicating in vivo and in vitro research.

4.  In vitro studies: add more detail if available in the literature

Answer: Details which was available in the literature were added to manuscript.

Reviewer 2 Report

Comments and Suggestions for Authors

Dear authors:

The work presented is a superficial summary of the published works of Rhododendron subsect. Ledum. This work is a necessary summary to know the state of the topic on which you wish to work, but this article does not present original results obtained according to scientific methodology, which not constitute an advance in the knowledge of the phytochemical profile and biological activity of said species.

It is a preliminary document to begin an investigation with a specific objective on the profile and activity of the species considered.

In my opinion it does not meet the scientific rigor to be published in this journal.

All the best.

Author Response

We are pleased to resubmit for publication the revised version of the manuscript “Phytochemical Profiling and Biological Activities of Rhododendron Subsect. Ledum: Discovering the Medicinal Potential of Labrador Tea Species in the Northern Hemisphere” (Manuscript ID:  plants-2858667). We have appreciated the reviewer 2 comments and have addressed each of their concerns as outlined below.

Reviewer 2: Dear authors:

The work presented is a superficial summary of the published works of Rhododendron subsect. Ledum. This work is a necessary summary to know the state of the topic on which you wish to work, but this article does not present original results obtained according to scientific methodology, which not constitute an advance in the knowledge of the phytochemical profile and biological activity of said species.

It is a preliminary document to begin an investigation with a specific objective on the profile and activity of the species considered.

In my opinion it does not meet the scientific rigor to be published in this journal.

All the best.

Answer: Thank you for the review. We appreciate your feedback and would like to address the concerns you have raised. Our current review work provides insights into the existing literature on the Rhododendron subsects. Ledum. The main scientific uncertainties lie within the intermix of the Rhododendron subsection Ledum nomenclature that includes various names. Our main goal was to identify the accepted species and to clarify the scientific data present on each accepted species. Coupling the proper plant name with its phytochemical profile, chemotype, chemical marker and activity is prerequisite knowledge for consistent future studies. Most research up to date has been devoted to the volatile fraction of the plant. However, phenolic and triterpene composition has been determined only for species grown in Canada and China. The research gaps within the phytochemical composition and present trends in the experimentally based biological activities will encourage future studies of subsection Ledum species towards the fields of ecology, medicine, materials science, and agriculture, making it a multi-disciplinary field with significant implications for solving scientific uncertainties and practical applications.

Reviewer 3 Report

Comments and Suggestions for Authors

General remarks. The authors performed a detailed analysis of the scientific literature and provided information on the nomenclature of the genus Rhododendron (Ledum), the distribution of plants of this genus in natural habitats, phytochemical composition, and biological activity. The analyzed 57 literature sources provide valuable information about the studies that have already been carried out.

The Abstract is clear and presents the results of the investigations.

The Introduction substantiates the aim of this review.

I missed the discussion of biological properties in the results section. I would recommend supplementing this section with at least a short description of the biological properties of the species (R. tomentosum and R. groenlandicum) "main species with ethnopharmacological data" (line 158). Especially since the abstract states that "review will comprehensively analyze...and biological properties" (lines 13-14).

The list of references is satisfactory. However, I recommend including the following articles as well:

https://doi.org/10.3390/molecules26237121

https://doi.org/10.3390/molecules23092285

https://doi.org/10.3390/molecules25071676

In my opinion, the conclusions are narrative. In the conclusions, I would suggest more emphasis on what research has already been carried out and what continuation of research the authors suggest, taking into account this review.

The list of references is satisfactory however, I recommend including these publications as well:

https://doi.org/10.3390/molecules26237121

https://doi.org/10.3390/molecules23092285

https://doi.org/10.3390/molecules25071676

Minor comments

Some remarks about nomenclature. One of the strengths of this review is that the authors chose to analyze existing studies on the genera Rhododendron and Ledum. There is much confusion in the systematics of these genera and species. The publications reviewed include published studies on Ledum palustre, which is a synonym of Rhododendron tomentosum. Therefore, it is very important to clarify species names everywhere in the text. Now R. tormentosum (lines 50, 52)  elsewhere - R. tomentosum (lines 74, 79, 80 etc.). I recommend the authors review all species names and correct them.

Author Response

We are pleased to resubmit for publication the revised version of the manuscript “Phytochemical Profiling and Biological Activities of Rhododendron Subsect. Ledum: Discovering the Medicinal Potential of Labrador Tea Species in the Northern Hemisphere” (Manuscript ID:  plants-2858667). We have appreciated the reviewer 3 comments and have addressed their concerns as outlined below.

Reviewer 3: General remarks. The authors performed a detailed analysis of the scientific literature and provided information on the nomenclature of the genus Rhododendron (Ledum), the distribution of plants of this genus in natural habitats, phytochemical composition, and biological activity. The analyzed 57 literature sources provide valuable information about the studies that have already been carried out.

The Abstract is clear and presents the results of the investigations.

The Introduction substantiates the aim of this review.

I missed the discussion of biological properties in the results section. I would recommend supplementing this section with at least a short description of the biological properties of the species (R. tomentosum and R. groenlandicum) "main species with ethnopharmacological data" (line 158). Especially since the abstract states that "review will comprehensively analyze...and biological properties" (lines 13-14).

Answer: Thank you very much for the suggestion. We have supplemented the section with the ethnopharmacological data available. We have elucidated the scientific uncertainties and coupled present chemotype information with the traditional medicinal data. Our main goal was to identify the accepted species and to clarify the scientific data present on each accepted species.

The list of references is satisfactory. However, I recommend including the following articles as well:

https://doi.org/10.3390/molecules26237121

https://doi.org/10.3390/molecules23092285

https://doi.org/10.3390/molecules25071676

Answer: Thank you for the suggestions. The references were included in  the manuscript.

In my opinion, the conclusions are narrative. In the conclusions, I would suggest more emphasis on what research has already been carried out and what continuation of research the authors suggest, taking into account this review.

Answer: Thank you for your constructive feedback. The conclusions were rewritten.

Minor comments

Some remarks about nomenclature. One of the strengths of this review is that the authors chose to analyze existing studies on the genera Rhododendron and Ledum. There is much confusion in the systematics of these genera and species. The publications reviewed include published studies on Ledum palustre, which is a synonym of Rhododendron tomentosum. Therefore, it is very important to clarify species names everywhere in the text. Now R. tormentosum (lines 50, 52)  elsewhere - R. tomentosum (lines 74, 79, 80 etc.). I recommend the authors review all species names and correct them.

Answer: All article species names were reviewed and corrected.